# LEARNING CURVES FOR ANALYSIS OF DEEP NETWORKS

## ABSTRACT

A *learning curve* models a classifier's test error as a function of the number of training samples. Prior works show that learning curves can be used to select model parameters and extrapolate performance. We investigate how to use learning curves to analyze the impact of design choices, such as pretraining, architecture, and data augmentation. We propose a method to robustly estimate learning curves, abstract their parameters into error and data-reliance, and evaluate the effectiveness of different parameterizations. We also provide several interesting observations based on learning curves for a variety of image classification models.

## 1 INTRODUCTION

What gets measured gets optimized. We need better measures of learning ability to design better classifiers and predict the payoff of collecting more data. Currently, classifiers are evaluated and compared by measuring performance on one or more datasets according to a fixed train/test split. Ablation studies help evaluate the impact of design decisions. However, one of the most important characteristics of a classifier, how it performs with varying numbers of training samples, is rarely measured or modeled.

In this paper, we refine the idea of *learning curves* that model error as a function of training set size. Learning curves were introduced nearly thirty years (e.g. by Cortes et al. (1993)) to accelerate model selection of deep networks. Recent works have demonstrated the predictability of performance improvements with more data (Hestness et al., 2017; Johnson & Nguyen, 2017; Kaplan et al., 2020; Rosenfeld et al., 2020) or more network parameters (Kaplan et al., 2020; Rosenfeld et al., 2020). But such studies have typically required large-scale experiments that are outside the computational budgets of many research groups, and their purpose is extrapolation rather than validating design choices. We find that a generalized power law function provides the best learning curve fit, while a model linear in $n^{-0.5}$, where $n$ is the number of training samples (or "training size"), provides a good local approximation. We abstract the curve into two key parameters: $e_N$ and $\beta_N$. $e_N$ is error at $n = N$, and $\beta_N$ is a measure of data-reliance, revealing how much a classifier's error will change if the training set size changes. Learning curves provide valuable insights that cannot be obtained by single-point comparisons of performance. Our aim is to promote the use of learning curves as part of a standard learning system evaluation.

Our **key contributions**:

- Investigate how to best model, estimate, characterize, and display learning curves for use in classifier analysis

- Use learning curves to analyze impact on error and data-reliance due to network architecture, depth, width, fine-tuning, data augmentation, and pretraining

Table 1 shows validated and rejected popular beliefs that single-point comparisons often overlook. In the following sections, we investigate how to model learning curves (Sec. 2), how to estimate them (Sec. 3), and what they can tell us about the impact of design decisions (Sec. 4), with discussion of limitations and future work in Sec. 5.

| Popular beliefs | Your guess | Our guess | Our result | Result figures |
|---|---|---|---|---|
| *Pre-training* on similar domains nearly always helps compared to training from scratch. | ☐ | T | T | 9a, 9b, 3 |
| *Pre-training*, even on similar domains, introduces bias that would harm performance with a large enough training set. | ☐ | T | ? | 3 |
| *Self-/un-supervised training* performs better than supervised pre-training for small datasets. | ☐ | T | F | 3 |
| *Fine-tuning* the entire network (vs. just the classification layer) is only helpful if the training set is large. | ☐ | F | F | 9a, 9b |
| *Increasing network depth*, when fine-tuning, harms performance for small training sets, due to an overly complex model. | ☐ | T | F | 4a, 4b |
| *Increasing network depth*, when fine-tuning, is more helpful for larger training sets than smaller ones. | ☐ | T | F | 4a, 4b |
| *Increasing network depth*, if the backbone is frozen, is more helpful for smaller training sets than larger ones. | ☐ | T | F | 4a, 4b |
| *Increasing depth or width* improves more than ensembles of smaller networks with the same number of parameters. | ☐ | T | T | 4e, 4a |
| *Data augmentation* is roughly equivalent to using a $m$-times larger training set for some $m$. | ☐ | T | T | 5 |

Table 1: **Deep learning quiz!** We encourage our readers to judge each claim as T (true) or F (false), and then compare to our guesses and results. In the results column, "T" means the experiments are consistent with the belief, "F" for inconsistent, and "?" for hard to say.

## 2 MODELING LEARNING CURVES

The learning curve measures test error $e_{test}$ as a function of the number of training samples $n$ for a given classification model and learning method. Previous empirical observations suggest a functional form $e_{test}(n) = \alpha + \eta n^{\gamma}$, with bias-variance trade-off and generalization theories typically indicating $\gamma = -0.5$. We summarize what bias-variance trade-off and generalization theories (Sec. 2.1) and empirical studies (Sec. 2.2) can tell us about learning curves, and describe our proposed abstraction in Sec. 2.3.

### 2.1 BIAS-VARIANCE TRADE-OFF AND GENERALIZATION THEORY

The bias-variance trade-off is an intuitive and theoretical way to think about generalization. The "bias" is error due to inability of the classifier to encode the optimal decision function, and the "variance" is error due to variations in predictions due to limited availability of training samples for parameter estimation. This is called a trade-off because a classifier with more parameters tends to have less bias but higher variance. Geman et al. (1992) decompose mean squared regression error into bias and variance and explore the implications for neural networks, leading to the conclusion that "identifying the right preconditions is *the* substantial problem in neural modeling". This conclusion foreshadows the importance of pretraining, though Geman et al. thought the preconditions must be built in rather than learned. Domingos (2000) extends the analysis to classification. Theoretically, the mean squared error (MSE) can be modeled as $e_{test}^2(n) = bias^2 + noise^2 + var(n)$, where "noise" is irreducible error due to non-unique mapping from inputs to labels, and variance can be modeled as $var(n) = \sigma^2/n$ for $n$ training samples.

The $\eta n^{-0.5}$ term appears throughout machine learning generalization theory. For example, the bounds based on hypothesis VC-dimension (Vapnik & Chervonenkis, 1971) and Rademacher Complexity (Gnecco & Sanguineti, 2008) are both $O(cn^{-0.5})$ where $c$ depends on the complexity of the classification model. More recent work also follows this form. We give some examples of bounds in Table 2 without describing all of the parameters because the point is that the test error bounds vary with training size $n$ as a function of $n^{-0.5}$, for all approaches.

Table 2: **Generalization bound examples:** The bounds each predict generalization error increasing as a function of $n^{-0.5}$. Note: variable notation is consistent only within each line, except $n$.

| Work | Key Variables | Bound |
|---|---|---|
| Neyshabur et al. (2018) | network depth $d$ | $O\left(n^{-0.5}Bd\sqrt{h\ln(dh)\pi_{i=1}^{d}\|\|W_i\|\|_2^2\sum_{i=1}^{d}\frac{\|\|W_i\|\|_F^2}{\|\|W_i\|\|_2^2}}/\gamma\right)$ |
| Bartlett et al. (2017) | spectral complexity $R_w$ | $O\left(\frac{\|\|X\|\|_2 R_w}{\gamma \cdot n}\log(\max_i h_i) + n^{-0.5}\right)$ |
| Arora et al. (2018) | compressibility to $q$ parameters with $r$ discrete values | $O\left(n^{-0.5}\sqrt{q\log r}\right)$ |
| Bousquet & Elisseeff (2002) | based on analysis of stability with margin $\gamma$ | $O\left(n^{-0.5}/\gamma\right)$ |

## 2.2 EMPIRICAL STUDIES

Some recent empirical studies (e.g. Sun et al. (2017)) claim a log-linear relationship between error and training size, but this holds only when asymptotic error is zero. Hestness et al. (2017) model error as $e_{test}(n) = \alpha + \eta n^\gamma$ but often find $\gamma$ much smaller in magnitude than $-0.5$ and suggest that poor fits indicate need for better hyperparameter tuning. This raises an interesting point that sample efficiency depends both on the classification model and on the efficacy of the optimization algorithm and parameters. Johnson & Nguyen (2017) also find a better fit with this extended power law model than by restricting $\gamma = -0.5$ or $\alpha = 0$. We find that, by selecting the learning rate through validation on one training size and using the Ranger optimizer (Wright, 2019), we can achieve a good approximate fit with $\gamma = -0.5$ and best fit with $-0.3 < \gamma < -0.7$.

In the language domain, learning curves are used in a fascinating study by Kaplan et al. (2020). For natural language transformers, they show that a power law relationship between logistic loss, model size, compute time, and dataset size is maintained if (and only if) each is increased in tandem. We draw some similar conclusions to their study, such as that increasing model size tends to improve performance *especially* for small training sets (which surprised us). However, the studies are largely complementary, as we study convolutional nets in computer vision, classification error (instead of logistic loss), and a broader range of design choices such as effects across depth, width, data augmentation, pretraining source, architecture, and dataset. Also related, Rosenfeld et al. (2020) model error as a function of both training size and number of model parameters with a five-parameter function that accounts for training size, model parameter size, and chance performance.

A key difference in our work is that we focus on how to best draw insights about design choices from learning curves, rather than on extrapolation. As such, we propose methods to estimate learning curves and their variance from a relatively small number of trained models.

## 2.3 PROPOSED CHARACTERIZATION OF LEARNING CURVES FOR EVALUATION

A classifier's performance can be characterized in terms of its error and data-reliance, or how quickly the error changes with training size $n$. With $e(n) = \alpha + \eta n^\gamma$, we find that $\gamma = -0.5$ provides a good local approximation but that fitting $\gamma$ significantly improves leave-one-size-out RMS error and extrapolation accuracy, as we detail in Sec. 4. However, $\alpha$, $\eta$, and $\gamma$ cannot be meaningfully compared across curves because the parameters have high covariance with small data perturbations, and comparing $\eta$ values is not meaningful unless $\gamma$ is fixed and vice-versa.

We propose to report error and sensitivity to training size in a way that can be derived from various learning curve models and is insensitive to data perturbations. The curve is characterized by error $e_N = \alpha + \eta N^\gamma$ and data-reliance $\beta_N$ at $N$, and we typically choose $N$ as the full dataset size. Noting that most learning curves are locally well approximated by a model linear in $n^{-0.5}$, we compute data-reliance as $\beta_N = N^{-0.5} \frac{\partial e}{\partial n^{-0.5}}\big|_{n=N} = -2\eta\gamma N^\gamma$. When the error is plotted against $n^{-0.5}$, $\beta_N$ is the slope at $N$ scaled by $N^{-0.5}$, where the scaling was chosen to make the practical implications of $\beta_N$ more intuitive. This yields a simple predictor for error when changing training size by a factor of $d$:

$$e(d \cdot N) = e_N + \left(\frac{1}{\sqrt{d}} - 1\right)\beta_N. \tag{1}$$

Thus, by this linearized estimate, asymptotic error is $e_N - \beta_N$, a 4-fold increase in data (*e.g.* $400 \to 1600$) reduces error by $0.5\beta_N$, and using only one quarter of the dataset (*e.g.* $400 \to 100$) increases the error by $\beta_N$. For two models with similar $e_N$, the one with a larger $\beta_N$ would outperform with more data but underperform with less. Note that $(e_N, \beta_N, \gamma)$ is a complete re-parameterization of the extended power law, with $\gamma + 0.5$ indicating the curvature in $n^{-0.5}$ scale.

## 3 ESTIMATING LEARNING CURVES

We now describe the method for estimating the learning curve from error measurements with confidence bounds on the estimate. Let $e_{ij}$ denote the random variable corresponding to test error when the model is trained on the $j^{\text{th}}$ fold of $n_i$ samples (either per class or in total). We assume $\{e_{ij}\}_{j=1}^{F_i}$ are i.i.d according to $\mathcal{N}(\mu_i, \sigma_i^2)$. We want to estimate learning curve parameters $\alpha$ (asymptotic error), $\eta$, and $\gamma$, such that $e_{ij} = \alpha + \eta n_i^\gamma + \epsilon_{ij}$ where $\epsilon_{ij} \sim \mathcal{N}(0, \sigma_i^2)$ and $\mu_{ij} = \mathbb{E}[e_{ij}] = \mu_i$.

Sections 3.1 and 3.2 describe how to estimate mean and variance of $\alpha$ and $\eta$ for a given $\gamma$, and Sec. 3.3 describes our approach for estimating $\gamma$.

## 3.1 WEIGHTED LEAST SQUARES FORMULATION

We estimate learning curve parameters $\{\alpha, \eta\}$ by optimizing a weighted least squares objective:

$$\mathcal{G}(\gamma) = \min_{\alpha, \eta} \sum_{i=1}^{S} \sum_{j=1}^{F_i} w_{ij} (e_{ij} - \alpha - \eta n^\gamma)^2 \tag{2}$$

where $w_{ij} = 1/(F_i \sigma_i^2)$. $F_i$ is the number of models trained with data size $n_i$ and is used to normalize the weight so that the total weight for observations from each training size does not depend on $F_i$. The factor of $\sigma_i^2$ accounts for the variance of $\epsilon_{ij}$. Assuming constant $\sigma_i^2$ and removing the $F_i$ factor would yield unweighted least squares.

The variance of the estimate of $\sigma_i^2$ from $F_i$ samples is $2\sigma_i^4/F_i$, which can lead to over- or under-weighting data for particular $i$ if $F_i$ is small. Recall that each sample $e_{ij}$ requires training an entire model, so $F_i$ is always small in our experiments. We would expect the variance to have the form $\sigma_i^2 = \sigma_0^2 + \hat{\sigma}^2/n_i$, where $\sigma_0^2$ is the variance due to random initialization and optimization and $\hat{\sigma}^2/n_i$ is the variance due to randomness in selecting $n_i$ samples. Indeed, by averaging over the variance estimates for many different network models on the CIFAR-100 (Krizhevsky, 2012) dataset, we find a good fit with $\sigma_0^2 = 0.2$. This enables us to estimate a single $\hat{\sigma}^2$ parameter from all samples $\mathbf{e}$ in a given learning curve as a least squares fit and also upper-bounds $w_{ij} <= 5$ even if two models happen to have the same error. This attention to $w_{ij}$ may seem fussy, but without such care we find that the learning curve often fails to account sufficiently for all the data in some cases.

## 3.2 SOLVING FOR LEARNING CURVE MEAN AND VARIANCE

Concatenating errors across dataset sizes (indexed by $i$) and folds results in an error vector $\mathbf{e}$ of dimension $D = \sum_{i=1}^{S} F_i$. For each $d \in \{1, \cdots, D\}$, $\mathbf{e}[d]$ is an observation of error at dataset size $n_{i_d}$ that follows $\mathcal{N}(\mu_{i_d}, \sigma_{i_d}^2)$ with $i_d$ mapping $d$ to the corresponding $i$.

The weighted least squares problem can be formulated as solving a system of linear equations denoted by $W^{1/2}\mathbf{e} = W^{1/2}A\boldsymbol{\theta}$, where $W \in \mathbb{R}^{D \times D}$ is a diagonal matrix of weights $W_{dd} = w_d$, $A \in \mathbb{R}^{D \times 2}$ is a matrix with $A[d,:] = [1 \ n_d^\gamma]$, and $\boldsymbol{\theta} = [\alpha \ \eta]^T$ are the parameters of the learning curve, treating $\gamma$ as fixed for now. The estimator for the learning curve is then given by $\hat{\boldsymbol{\theta}} = (W^{1/2}A)^+ W^{1/2}\mathbf{e} = M\mathbf{e}$, where $M \in \mathbb{R}^{2 \times D}$ and $+$ is pseudo-inverse operator.

We compute a mean curve using

$$\overline{\boldsymbol{\theta}} = \mathbb{E}[\hat{\boldsymbol{\theta}}] = M\mathbb{E}[\mathbf{e}] = M\boldsymbol{\mu} \tag{3}$$

where $\boldsymbol{\mu} \in \mathbb{R}^D$ with $\boldsymbol{\mu}[d] = \hat{\mu}_{i_d}$ computed by empirical estimate as $\sum_{j=1}^{F_i} e_{ij}/F_i$.

The covariance of the estimator is given by

$$\Sigma_{\hat{\boldsymbol{\theta}}} = M\Sigma_{\mathbf{e}}M^T \tag{4}$$

where $\Sigma_{\hat{\boldsymbol{\theta}}} \in \mathbb{R}^{2 \times 2}$ and $\Sigma_{\mathbf{e}} \in \mathbb{R}^{D \times D}$ is the covariance of $\mathbf{e}$, where $\Sigma_{\mathbf{e}}[d_1, d_2] = \sigma_{i_{d_1}}^2$ if $i_{d_1} = i_{d_2}$ and 0 otherwise. We compute our empirical estimate of $\sigma_i^2$ as described in Sec. 3.1.

Since the curve is given by $e(n) = [1 \ n^\gamma]\boldsymbol{\theta}$, the mean curve can be computed as

$$\overline{e}(n) = [1 \ \ n^\gamma]\overline{\boldsymbol{\theta}} = \overline{\alpha} + \overline{\eta}n^\gamma. \tag{5}$$

The 95% bounds at any $n$ can be computed as $\overline{e}(n) \pm 1.96 \times \hat{\sigma}(n)$ with

$$\hat{\sigma}^2(n) = [1 \ \ n^\gamma]\Sigma_{\hat{\boldsymbol{\theta}}} \begin{bmatrix} 1 \\ n^\gamma \end{bmatrix} \tag{6}$$

where $\hat{\alpha}$ and $\hat{\eta}$ are the empirical estimates of $\alpha$ and $\eta$. These confidence bounds reflect the variance in error measurements, assuming the parameterization is capable of fitting the true mean.

### 3.3 ESTIMATING $\gamma$

We search for $\gamma$ that minimizes the weighted least squares objective with an L1-prior that slightly encourages values close to $0.5$. Specifically, we solve

$$\min_{\gamma \in (-1,0)} \mathcal{G}(\gamma) + \lambda|\gamma + 0.5| \tag{7}$$

by searching over $\gamma \in \{-0.99, ..., -0.01\}$ with $\lambda = 5$ for our experiments.

## 4 EXPERIMENTS

We describe our implementation details in Sec. 4.1, apply learning curves to gain insights about error and data-reliance in Sec. 4.2, and validate our choice of learning curve parameterization and fitting weights used in the least squares objective in Sec. 4.3.

### 4.1 IMPLEMENTATION DETAILS

We use Pytorch-Lightning (Falcon, 2019) for our implementation with various architectures, weight initializations, data augmentation, and linear or fine-tuning optimization.

**Training:** We train models with images of size $32 \times 32$ for CIFAR (Krizhevsky, 2012) and $224 \times 224$ for Places365 (Zhou et al., 2017) with a batch size of 64 (except for Wide-ResNet101 and Wide-ResNeXt101, where we use a batch size of 32 and performed one optimizer step every two batches). For each experiment setting, we conduct a learning rate search on a subset of the training data and choose the learning rate with the highest validation accuracy, and use it for all other subsets. We determine each fold's training schedule on a mini-train/mini-val split of 2:1 on the train set. Each time the mini-val error stops decreasing for some epochs ("patience"), we revert to the best epoch and decrease the learning rate to 10%, and we perform this twice. Then we use this optimal mini-train learning rate schedule and ending epoch to train on the whole fold. The patience is $\propto 1/\sqrt{n}$, and is 5 at the $n = 400$ samples/class for CIFAR100/Places365 and 15 at the largest training size for other smaller datasets. We use a weight decay value of 0.0001. We use the Ranger optimizer (Wright, 2019), which combines Rectified Adam (Liu et al., 2019), Look Ahead (Zhang et al., 2019), and Gradient Centralization (Yong et al., 2020). In early experiments, we found Ranger to lead to lower error and to reduce sensitivity of hyperparameters, compared to vanilla SGD or Adam (Kingma & Ba, 2015).

**Backbone Architecture:** We use the default Pytorch implementations of all of the following architectures: AlexNet (Krizhevsky et al., 2012), ResNet-18, ResNet-50, ResNet-101 (He et al., 2015b), ResNeXt-50, ResNeXt-100 (Xie et al., 2016), VGG16_BN (Simonyan & Zisserman, 2014), Wide-ResNet-50, and Wide-ResNet-101 (Zagoruyko & Komodakis, 2016). For each architecture, we modify the last layer to match the same number of classes as the test dataset with Kaiming initialization (He et al., 2015a).

**Number of Training Examples:** To compute learning curves for CIFAR and Places365, we vary the number of training examples per class, partition the train set, and train one model per partition. For CIFAR100 (Krizhevsky, 2012), we use $\{25, 50, 100, 200, 400\}$ training examples per class, and the number of models trained for each respectively is $\{16, 8, 4, 2, 1\}$. Similar to Hestness et al. (2017), we find training sizes smaller than 25 samples per class are strongly influenced by bounded error and deviate from our model. For Places365 dataset, we use $\{25, 50, 100, 200, 400, 1600\}$ training examples per class and $\{16, 8, 4, 3, 3, 1\}$ models each. For other datasets (Fig. 6), we use $\{20\%, 40\%, 80\%\}$ of the full data and train $\{4, 2, 1\}$ models each. We hold out 20% of data from the original training set for testing (a validation set could also be used if available) to discourage meta-fitting on the test set. For example, we hold out 100 samples per class from the original CIFAR100 training set and perform hyperparameter selection and training on the remaining 400 samples.

**Pretraining:** When pretraining is used, we initialize models with pretrained weights learned through supervised training on ImageNet or Places365, or MOCO self-supervised training on ImageNet (He et al., 2020). Otherwise, weights are randomly initialized with Kaiming initialization.

**Data Augmentation:** For CIFAR, we pad by 4 pixels and use a random $32 \times 32$ crop (test without augmentation), and for Places365 we use random-sized crop (Szegedy et al., 2015) to $224 \times 224$

and random flipping (center crop 224×224 test time). For remaining datasets, we follow the pre-processing in Zhai et al. (2020) that produced the best results when training from scratch.

**Linear vs. Fine-tuning:** For "linear", we only train the final classification layer, with the other weights frozen to initialized values. All weights are trained when "fine-tuning".

## 4.2 LEARNING CURVE COMPARISONS

We plot the fitted learning curves and confidence bounds, with observed test errors as black points. The legend displays $\gamma$, error $e_N$, and data reliance $\beta_N$ with $N = 400$. The x-axis is in scale $n^{-0.5}$ ($n$ in parentheses), but best-fitting $\gamma$ is used in all cases. In all plots, $n$ denotes number of samples per class except Fig. 6 where $n$ is the total number of samples. A vertical bar indicates $n = 1600$, which we consider the limit of accurate extrapolation from curves fit to $n \leq 400$ samples. All points are used for fitting, except in Fig. 9b $n = 1600$ is held out to test extrapolation.

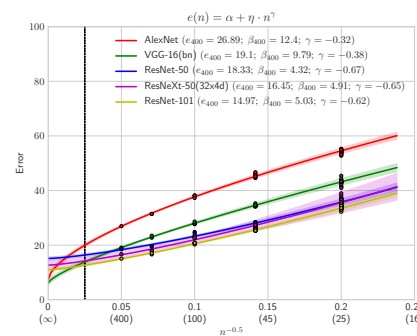

Figure 1: **Architecture** (w/ finetuning)

**Network architecture:** Advances in CNN architectures have reduced number of parameters while also reducing error over the range of training sizes. On CIFAR100, AlexNet has 61M parameters; VGG-16, 138M; ResNet-50, 26M; ResNeXt-50, 25M; and ResNet-101, 45M. Fig. 1 shows that each major advance through ResNet reduces both data-reliance and $e_{400}$, while ResNeXt appears to slightly reduce $e_{400}$ without change to data-reliance.

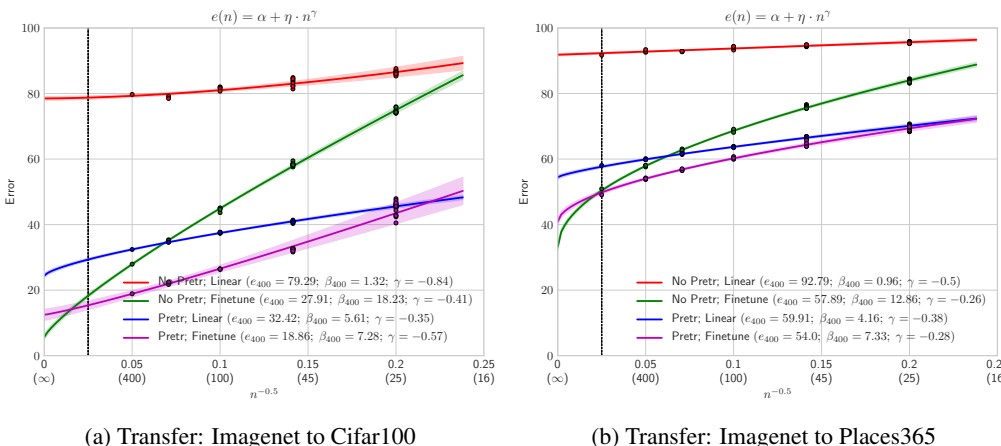

(a) Transfer: Imagenet to Cifar100

(b) Transfer: Imagenet to Places365

Figure 2: **Pretraining and fine-tuning** with ResNet-18.

**Pretraining and fine-tuning:** In Fig. 2 we see that, for linear classifiers, pretraining leads to a huge improvement in $e_{400}$ though with a moderate increase in data-reliance. When fine-tuning, the pretraining greatly reduces data-reliance $\beta_{400}$ and also reduces $e_{400}$. Pretraining clearly improves performance with smaller training sizes. However, we cannot draw conclusions about bias because extrapolated asymptotic error is not reliable, and the full story is complicated. On an object detection task, He et al. (2019) find that, with long learning schedules, randomly initialized networks approach the performance of pretrained networks (for the CNN backbone), even with finite data. Experiments by Zoph et al. (2020), also on object detection, show that pretraining can sometimes harm performance when strong data augmentation is used. Ko-

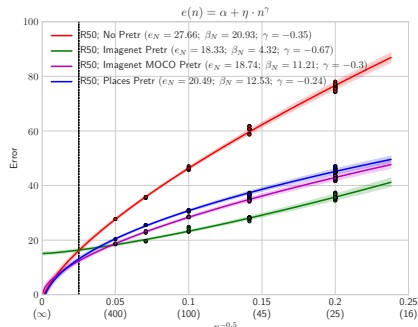

Figure 3: **Pretraining sources** (test on Cifar100).

rnblith et al. (2019) show that fine-tuned pretrained models outperform randomly initialized models on many datasets, but the gap is often small and narrows as data size grows. All agree that pretraining is at least important for providing a warm start that greatly reduces the training time, but whether it introduces bias (i.e. asymptotic error) likely depends on the tasks, domains, and optimization settings.

**Pretraining data sources:**  In Fig. 3, we compare pretraining strategies: random, supervised on ImageNet or Places365, and self-supervised on ImageNet (MOCO by He et al. (2020)). All initializations have similar extrapolated error at $n = 1600$, but different data-reliance. Self-supervised MOCO leads to lower $e_{400}$ and $\beta_{400}$ compared to Places365 pretraining. Supervised pretraining on ImageNet has the lowest $e_{400}$ and $\beta_{400}$. We suspect that the $\gamma = -0.67$ and higher extrapolated asymptotic error may be due to measurement noise and suboptimal hyperparameter selection.

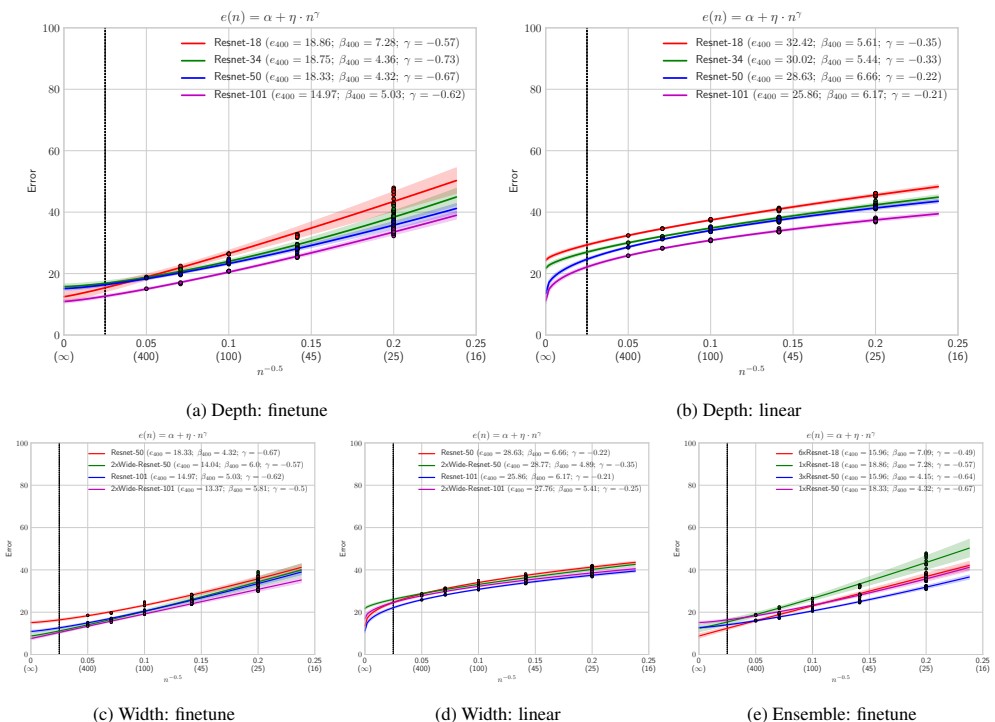

Figure 4: **Depth, width, and ensembles** on Cifar100.

**Network depth, width, and ensembles:**  The classical view is that smaller datasets need simpler models to avoid overfitting. In Figs. 4a, 4b, we show that, not only do deeper networks have better potential at higher data sizes, their data reliance does not increase (nearly parallel and drops a little for fine-tuning), making deeper networks perfectly suitable for smaller datasets. For linear classifiers (Fig. 4b), the deeper networks provide better features, leading to consistent drop in $e_{400}$. The small jump in data reliance between Resnet-34 and Resnet-50 may be due to the increased last layer input size from 512 to 2048 nodes. When increasing width, the fine-tuned networks (Fig. 4c) have reduced $e_{400}$ without much change to data-reliance. With linear classifiers (Fig. 4d), increasing the width leads to little change or even increase in $e_{400}$ with slight decrease in data-reliance. Rosenfeld et al. (2020) show that error can be modeled as a function of either training size, model size, or both. Modeling both jointly can provide additional capabilities such as selecting model size based on data size, but requires many more experiments to fit the curve.

An alternative to using a deeper or wider network is forming ensembles. In Figure 4e, we find that while an ensemble of six ResNet-18's (each 11.7M parameters) improves over a single model, it has higher $e_{400}$ and data-reliance than ResNet-101 (44.5M), Wide-ResNet-50 (68.9M), and Wide-ResNet-101 (126.9M). Three ResNet-50's (each 25.6M) underperforms Wide-ResNet-50 on $e_{400}$ but outperforms for small amounts of data due to lower data reliance.

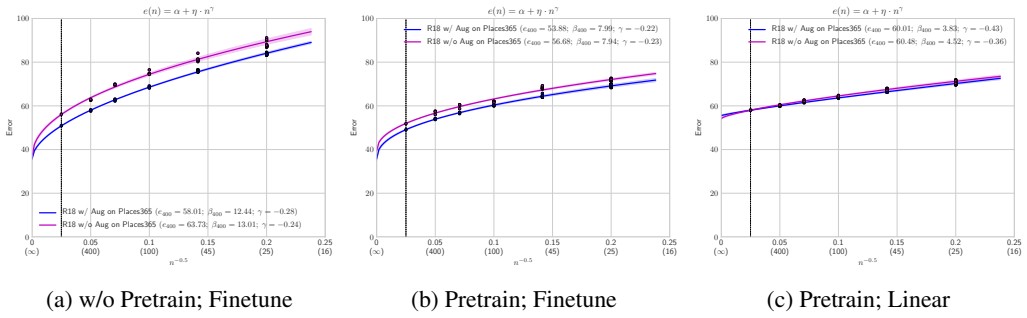

(a) w/o Pretrain; Finetune     (b) Pretrain; Finetune     (c) Pretrain; Linear

Figure 5: **Data augmentation** on Places365.

**Data Augmentation:** One may expect that data augmentation acts as a regularizer with reduced effect for large training sizes, or even possibly negative effect due to introducing bias. However, Fig. 5 shows that data augmentation on Places365 reduces error for all training sizes with little or no change to data-reliance when fine-tuning. $e(n)$ with augmentation roughly equals $e(1.8n)$ without it, supporting the view that augmentation acts as a multiplier on the value of an example. For the linear classifier, data augmentation has little apparent effect due to low data-reliance, but the results are still consistent with this multiplier.

**Additional datasets:** In Fig. 6, we verify that our learning curve model fits to multiple other datasets (chosen from natural tasks in Zhai et al. (2020)), comparing fine-tuned vs. linear with Resnet-18. For these plots only, $n$ is the total number of samples. The $\gamma$ values are estimated from data, but the prior has more effect here due to fewer error measurements.

We see fine-tuning consistently outperforms linear, though the difference is most dramatic for Sun397. Pretraining provides large benefits across datasets.

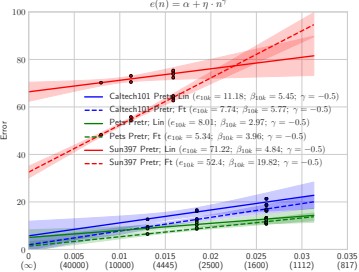

Figure 6: **Additional datasets**

### 4.3 EVALUATION OF LEARNING CURVES MODEL AND FITTING

We validate our learning curve model using leave-one-size-out prediction error, for example, predicting empirical mean performance with 400 samples per class based on observing error from models trained on 25, 50, 100, and 200 samples. We consider various choices of weighting schemes ($w$'s in Eq. 2) and estimating different parameters in a general form of the learning curve given by $e(n) = \alpha + \eta n^{\gamma} + \delta n^{2\gamma}$. Note that setting $\delta = 0$ yields the learning curve model described in Sec. 3.

**Weighting Schemes.** In the Fig. 7 table, we compare three weighting schemes across 16 classifiers: $w_{ij} = 1$ is unweighted; $w_{ij} = 1/\sigma_i^2$ is weighted by estimated size-dependent standard deviation; $w_{ij} = 1/(F_i \sigma_i^2)$ makes the total weight for a given dataset size invariant to the number of folds. On average our proposed weighting performs best with high significance compared to unweighted. The p-value is paired t-test of difference of means calculated across all dataset sizes.

**Model Choice.** We consider other parameterizations that are special cases of $e(n) = \alpha + \eta n^{\gamma} + \delta n^{2\gamma}$. The table in Fig. 7 shows that the parameterization used for our experiments outperforms the others, in most cases with high significance, and achieves a very good fit with $R^2$ of 0.998.

**Model Stability.** We test stability and sample requirements by repeatedly fitting curves to four resampled data points for a model (Resnet-18, no pretraining, fine-tuned, tested on Places365). Based on estimates of mean and standard deviation, one point each at $n = \{50, 100, 200, 400\}$ is sampled and used to fit a curve, repeated 100 times. Parentheses in legend show standard deviation of estimates of $e_N$, $\beta_N$, and $\gamma$. Our preferred model extrapolates best to $n = 1600$ and $n = 25$ while retaining stable estimates of of $e_N$ and $\beta_N$, but predicted asymptotic error $\alpha$ varies widely.

Appendix C shows similar estimates of $e_N$ and $\beta_N$ by fixing $\gamma = -0.5$ and fitting only $\alpha$ and $\eta$ on the three largest sizes (typically $n = \{100, 200, 400\}$), indicating that a lightweight approach of training a few models can yield similar conclusions.

| Params | Weights | $R^2$ | RMSE | | | | | | |
| --- | --- | --- | --- | --- | --- | --- | --- | --- | --- |
| | | | 25 | 50 | 100 | 200 | 400 | avg | p-value |
| $\alpha, \eta, \gamma$ | $\frac{1}{\sigma_i^2 F_i}$ | 0.998 | 2.40 | 0.86 | **0.54** | 0.57 | **0.85** | **1.04** | - |
| | $\frac{1}{\sigma_i^2}$ | 0.999 | **2.38** | 0.83 | 0.69 | 0.54 | 1.08 | 1.10 | 0.06 |
| | 1 | 0.998 | 2.66 | 0.86 | 0.79 | **0.50** | 1.26 | 1.21 | 0.008 |
| $\alpha, \eta$ | $\frac{1}{\sigma_i^2 F_i}$ | 0.988 | 3.41 | 1.09 | 0.69 | 0.72 | 1.21 | 1.42 | <0.001 |
| $\alpha, \eta, \delta$ | $\frac{1}{\sigma_i^2 F_i}$ | 0.999 | 2.89 | **0.74** | 0.68 | 0.56 | 0.94 | 1.16 | 0.05 |
| $\alpha, \eta, \delta, \gamma$ | $\frac{1}{\sigma_i^2 F_i}$ | 0.999 | 3.46 | **0.74** | 0.70 | 0.59 | 1.00 | 1.30 | 0.02 |

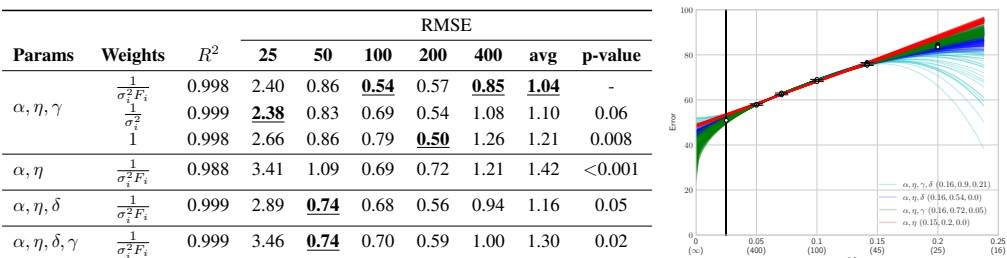

Figure 7: **Learning curve model and weights validation.** See text for explanation.

## 5 LIMITATIONS AND FUTURE WORK

**Limitations:** Our work in this paper is limited to classification loss, and our model does not account for small sample effects where chance performance is a major factor. Although our proposed $e_N$ and $\beta_N$ are stable under perturbations and different learning curve parameterizations, the asymptotic error $\alpha$ and exponent $\gamma$ parameters of the learning curve are unstable, and our confidence interval does not account for $\gamma$ variance. Unstable $\alpha$ means that little can be concluded about asymptotic performance, though $e_N - \beta_N$ can stand in as a measure of large-data performance. Unstable $\gamma$ may mean that conclusions are subject to the hyperparameter selection and optimization method.

**Future work:** Do the hyperparameters such as learning rate, schedule, and weight decay determine $\gamma$, or something else? It appears that $\gamma < -0.5$ is accompanied by high $\alpha$ and/or $\eta$. Should $\gamma = -0.5$ for a well-trained system? Answering these questions could lead to improved training and evaluation methodologies. It would also be interesting to investigate learning curve models for small training size, other losses and prediction types, more design parameters and interactions, and impact of imbalance in class distribution.

Appendix A offers extended discussion. Appendix B provides a guide to fitting, displaying, and using learning curves. Appendix C contains a table of learning curves for all of our experiments and compares $e_N$ and $\beta_N$ produced by two learning curve models.

## 6 CONCLUSION

We investigate learning curve models for analyzing classifier design decisions. We find an extended power law provides the best fit across many different architectures, datasets, and other design parameters. We propose to characterize error and data-reliance with $e_N$ and $\beta_N$, which are stable under data perturbations and can be derived from different learning curve models. Our experiments lead to several interesting observations about impacts of pretraining, fine-tuning, data augmentation, depth, width, and ensembles. We anticipate learning curves can further inform training methodology, continual learning, and representation learning, among other problems, and hope to see learning curves become part of a standard classification evaluation.

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

## A    EXTENDED DISCUSSION

Evaluation methodology is the foundation of research, impacting how we choose problems and rank solutions. Large train and test sets now serve as the fuel and crucible to refine machine learning methods. The current evaluation standard of using fixed i.i.d. train/test sets has supported many classification model improvements, but as machine learning broadens to continual learning, representation learning, long-tail learning, and so on, we need evaluation methods that better reflect the uncontrollable, unpredictable, and ever-changing world. By characterizing performance in terms

of error and data-reliance, we can provide a more complete understanding of model design and training size impact than single-point error. With that perspective, we discuss the limitations of our experiments and directions for future work.

- **Cause and impact of $\gamma$:** We speculate that $\gamma$ is largely determined by hyperparameters and optimization rather than model design, so conclusions are conditioned on particular training parameters. This presents an opportunity to identify poor training regimes and improve them. Intuitively, one would expect that more negative $\gamma$ values are better (i.e. $\gamma = -1$ preferable to $\gamma = -0.5$), since the curve is curve $O(n^\gamma)$, but we find the high-magnitude $\gamma$ tends to come with high asymptotic error, indicating that the efficiency comes at cost of over-commitment to initial conditions. We speculate (but with some disagreement among authors) that $\gamma = -0.5$ is an indication of a well-trained curve and will generally outperform curves with higher or lower $\gamma$, given the same classification model. It would be interesting to examine the impact of hyperparameter selection and optimization method on $\gamma$.

- **Small training sets**: Error is bounded and, with small training sets, classifier performance may be modeled as transitioning from random guess to informed prediction, as shown by Rosenfeld et al. (2020). We do not model performance with very small training size, partly to keep our model simple, partly because small training performance can be easily measured empirically, and partly because performance with small training size is highly variable depending on the sample. However, studying performance with small training sizes could be interesting, particularly to determine whether design decisions have an impact at the small size that is not apparent at larger sizes.

- **Losses and Prediction types**: We analyze multiclass classification error, but the same analysis could likely be extended to other losses and prediction types. For example, Kaplan et al. (2020) analyze learning manifolds of cross-entropy loss, which is unbounded, of language model transformers. Problems like object detection or grounding sometimes have relatively complex evaluation measures, such as average precision after accounting for localization and label accuracy, but test evaluation of the same losses used for training should still apply. Sun et al. (2017) show an approximately log-linear behavior between mean intersection of union semantic segmentation error as a function of number of training samples.

- **More design parameters and interactions**: The interaction between data scale, model scale, and performance is well-explored by Kaplan et al. (2020) and Rosenfeld et al. (2020), but it could also be interesting to explore interactions, e.g. between class of architecture (e.g. VGG, ResNet, EfficientNet (Tan & Le, 2019)) and some design parameters, to see how ideas such as skip-connections, residual layers and creating bottlenecks influence performance. More extensive evaluation of data augmentation, representation learning, optimization and regularization methods would also be interesting.

- **Unbalanced class distributions**: In most of our experiments, we use equal number of samples per class. Further experimentation is required to determine whether class imbalance impacts the form of the learning curve.

## B  USER'S GUIDE TO LEARNING CURVES

### B.1  USES FOR LEARNING CURVES

- **Comparison**: When comparing two learners, measuring the error and data-reliance provides a better understanding of the differences than evaluating single-point error. We compare curves with $e_N$ and $\beta_N$, rather than directly using the curve parameters, because they are more stable under data perturbations and do not depend on the parameterization, instead corresponding to error and rate of change about $n = N$. $e_N - \beta_N$ can be used as a measure of large-sample performance.

- **Performance extrapolation**: A 10x increase in training data can require a large investment, sometimes millions of dollars. Learning curves can predict how much performance will improve with the additional data to judge whether the investment is worthwhile.

- **Model selection**: When much training data is available, architecture, hyperparameters, and losses can be designed and selected using a small subset of the data to minimize the extrapolated error of the full training set size. Higher-parameter models such as in Kaplan et al. (2020) and Rosenfeld et al. (2020) may be more useful as a mechanism to simultaneously select scale parameters and extrapolate performance, though fitting those models is much more computationally expensive due to the requirement of sampling error/loss at multiple scales and data sizes.

- **Hyperparameter validation**: A poor fitting learning curve (or one with $\gamma$ far from $-0.5$) is an indication of poor choice of hyperparameters, as pointed out by Hestness et al. (2017).

### B.2 ESTIMATING AND DISPLAYING LEARNING CURVES

**Use validation set:** We recommend computing learning curves on a validation set, rather than a test set, according to best practice of performing a single evaluation on the test set for the final version of the algorithm. All of our experiments are on a validation set, which is carved from the official training set if necessary.

**Generate at least four data points:** In most of our experiments on CIFAR100, we train a 31 models: 1 on 400 images, 2 on 200 images, 4 on 100 images, 8 on 50 images, and 16 on 25 images. Each trained model provides one data point, the average validation error. In each case, the training data is partitioned so that the image sets within the same size are non-overlapping. Training multiple models at each size enables estimating the standard deviation for performing weighted least squares and producing confidence bounds. However, our experiments indicate that learning curves are highly stable, so a minimal experiment of training four models on the full, half, quarter, and eighth-size training set may be sufficient as part of a standard evaluation. See Fig. 8 It may be necessary to train more models if attempting to distinguish fine differences.

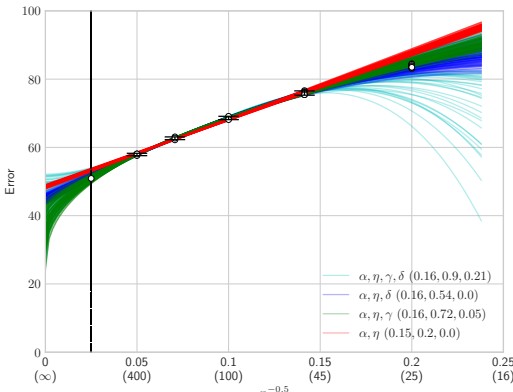

Figure 8: Stability under sparse measurements: Sampled learning curves for Places365 fine-tuned without pretraining are shown for four different learning curve parameterizations. In each case, means and standard deviations (shown by error bars) are estimated for $n = 50$, $n = 100$, $n = 200$, $n = 400$, using all the data points shown as white circles. Then, 100 times, we sample one point each from a Guassian distribution and fit a learning curve to the four points. In parantheses, the legend shows the standard deviation of $e_N$, $\beta_N$, and $\gamma$. Note that the parameterization of $\{\alpha, \eta, \gamma\}$ extrapolates best to lower and higher data sizes while still producing stable estimates of $e_N$ and $\beta_N$. Asymptotic error, however, varies widely.

**Set hyperparameters:** The learning rate and learning schedule are key parameters to be set. We have not experimented with changes to weight decay, momentum, or other hyperparameters.

**Fit learning curves:** If more than one data point is available for the same training size, the standard deviation can be estimated. As described in Sec. 3, we recommend fitting a model of $\sigma_i^2 = \sigma_0^2 + \hat{\sigma}^2/n$, where $\sigma_0^2$. $\sigma_0^2$ is the variance due to randomness in initialization and optimization. The fitting is not highly sensitive to this parameter, so we recommend setting $\sigma_0^2 = 0.01$ and fitting $\hat{\sigma}$ to observations, since estimating both from experiments to generate a single learning curve introduces high variance and instability.

**Display learning curves or parameters:** As in this paper, learning curves can be plotted linearly with the x-axis as $n^{-0.5}$ and the y-axis as error. We choose this rather than log-linear because it reveals prediction of asymptotic error and yields a linear plot when $\gamma = -0.5$. Since space is often a premium, the learning curve parameters can be displayed instead, as illustrated in Table 3.

Table 3: Results: $\mathrm{model}_1$ and $\mathrm{model}_2$ have similar percent test error when training on the full set. Fitting a learning curve on the validation set, we see that $\mathrm{model}_2$ has higher data-reliance, so may outperform for larger training sets. *This is a hypothetical example to illustrate use of learning curves in a table.*

|  | $e_N$ | $\beta_N$ | $\gamma$ |
|---|---|---|---|
| $\mathrm{model}_1$ | 25.3 % | 4.6 | -0.36 |
| $\mathrm{model}_2$ | 25.2 % | 8.4 | -0.47 |

## C   TABLE OF LEARNING CURVES

Table 4 shows experimental settings and fit parameters for learning curves under two parameterizations. We can see that similar $e_{400}$ and $\beta_{400}$ values are obtained when fixing $\gamma = -0.5$ and fitting to errors with only three training sizes (RMS difference in $e_{400}$ and $\gamma_{400}$ are 0.42 and 0.95, respectively). This means that learning curves can be fit and compared without training a large number of additional models.

Table 4: Experiment settings and parameters: We show the datasets, architectures, settings, and learning rate (set by mini-train/val) used to train and test our classifiers. Next, we show the parameters fit using the extended power law model $e(n) = \alpha + \eta n^{-\gamma}$. Next to that, we show the model resulting from setting $\gamma = -0.5$ and fitting to only the three training sizes with highest n.

| | dataset | arch | # param | pretrain/init | fine-tune? | data aug? | lrnRate | extended power law | | | | | $n^{-0.5}$ linear fit to last 3 points | | | |
|---|---|---|---|---|---|---|---|---|---|---|---|---|---|---|---|---|
| | | | | | | | | $\alpha$ | $\eta$ | $\gamma$ | $e_{400}$ | $\beta_{400}$ | $\alpha$ | $\eta$ | $e_{400}$ | $\beta_{400}$ |
| **PRETRAIN_IN2CIFAR** | | | | | | | | | | | | | | | | |
| No Pretr; Linear | CIFAR | Resnet-18 | 51K | Random | No | Yes | 0.01 | 78.51 | 120.13 | -0.84 | 79.29 | 1.32 | 78.06 | 26.12 | 79.36 | 1.31 |
| No Pretr; Finetune | CIFAR | Resnet-18 | 11.7M | Random | Yes | Yes | 0.01 | 5.68 | 259.29 | -0.41 | 27.91 | 18.23 | 11.21 | 336.13 | 28.02 | 16.81 |
| Pretr; Linear | CIFAR | Resnet-18 | 51K | ImageNet | No | Yes | 0.0003 | 24.4 | 65.28 | -0.35 | 32.42 | 5.61 | 27.33 | 102.16 | 32.44 | 5.11 |
| Pretr; Finetune | CIFAR | Resnet-18 | 11.7M | ImageNet | Yes | Yes | 0.001 | 12.48 | 194.19 | -0.57 | 18.86 | 7.28 | 11.37 | 150.73 | 18.91 | 7.54 |
| **PRETRAIN_IN2PLACES** | | | | | | | | | | | | | | | | |
| No Pretr; Linear | Places | Resnet-18 | 187K | Random | No | Yes | 0.03 | 91.84 | 19.13 | -0.5 | 92.79 | 0.96 | 91.09 | 29.31 | 92.55 | 1.47 |
| No Pretr; Finetune | Places | Resnet-18 | 11.7M | Random | Yes | Yes | 0.001 | 33.16 | 117.45 | -0.26 | 57.89 | 12.86 | 44.39 | 263.63 | 57.57 | 13.18 |
| Pretr; Linear | Places | Resnet-18 | 187K | ImageNet | No | Yes | 0.0003 | 54.43 | 53.39 | -0.38 | 59.91 | 4.16 | 56.11 | 76.95 | 59.95 | 3.85 |
| Pretr; Finetune | Places | Resnet-18 | 11.7M | ImageNet | Yes | Yes | 0.0003 | 40.92 | 70.04 | -0.28 | 54 | 7.33 | 44.82 | 174.69 | 53.55 | 8.73 |
| **PRETRAIN_IN_PLACES_MOCO2CIFAR** | | | | | | | | | | | | | | | | |
| No Pretr | CIFAR | Resnet-50 | 25.6M | Random | Yes | Yes | 0.01 | -2.23 | 243.44 | -0.35 | 27.66 | 20.93 | 9.18 | 372.11 | 27.79 | 18.61 |
| Pretr on Imagenet | CIFAR | Resnet-50 | 25.6M | ImageNet | Yes | Yes | 0.001 | 15.11 | 178.61 | -0.67 | 18.33 | 4.32 | 13.3 | 99.88 | 18.29 | 4.99 |
| Pretr on Places | CIFAR | Resnet-50 | 25.6M | Places | Yes | Yes | 0.001 | -5.61 | 109.92 | -0.24 | 20.49 | 12.53 | 9.05 | 195.43 | 18.82 | 9.77 |
| Pretr on Imagenet with MOCO | CIFAR | Resnet-50 | 25.6M | ImageNet (MOCO) | Yes | Yes | 0.0003 | 0.07 | 112.69 | -0.3 | 18.74 | 11.21 | 10.07 | 210.67 | 20.61 | 10.53 |
| **DEPTH_FT** | | | | | | | | | | | | | | | | |
| Resnet-18 | CIFAR | Resnet-18 | 11.7M | ImageNet | Yes | Yes | 0.001 | 12.48 | 194.19 | -0.57 | 18.86 | 7.28 | 11.37 | 150.73 | 18.91 | 7.54 |
| Resnet-34 | CIFAR | Resnet-34 | 21.8M | ImageNet | Yes | Yes | 0.001 | 15.76 | 237.19 | -0.73 | 18.75 | 4.36 | 13.51 | 104.15 | 18.72 | 5.21 |
| Resnet-50 | CIFAR | Resnet-50 | 25.6M | ImageNet | Yes | Yes | 0.001 | 15.11 | 178.61 | -0.67 | 18.33 | 4.32 | 13.3 | 99.88 | 18.29 | 4.99 |
| Resnet-101 | CIFAR | Resnet-101 | 44.5M | ImageNet | Yes | Yes | 0.0003 | 10.91 | 166.44 | -0.62 | 14.97 | 5.03 | 8.95 | 117.22 | 14.81 | 5.86 |
| **DEPTH_LINEAR** | | | | | | | | | | | | | | | | |
| Resnet-18 | CIFAR | Resnet-18 | 51K | ImageNet | No | Yes | 0.001 | 24.4 | 65.28 | -0.35 | 32.42 | 5.61 | 27.33 | 102.16 | 32.44 | 5.11 |
| Resnet-34 | CIFAR | Resnet-34 | 51K | ImageNet | No | Yes | 0.001 | 21.77 | 59.56 | -0.33 | 30.02 | 5.44 | 25.11 | 98.33 | 30.03 | 4.92 |
| Resnet-50 | CIFAR | Resnet-50 | 205K | ImageNet | No | Yes | 0.0003 | 13.5 | 56.54 | -0.22 | 28.63 | 6.66 | 23.05 | 112.08 | 28.65 | 5.6 |
| Resnet-101 | CIFAR | Resnet-101 | 205K | ImageNet | No | Yes | 0.0003 | 11.17 | 51.7 | -0.21 | 25.86 | 6.17 | 20.98 | 99.32 | 25.95 | 4.97 |
| **WIDTH_FT** | | | | | | | | | | | | | | | | |
| Resnet-50 | CIFAR | Resnet-50 | 25.6M | ImageNet | Yes | Yes | 0.001 | 15.11 | 178.61 | -0.67 | 18.33 | 4.32 | 13.3 | 99.88 | 18.29 | 4.99 |
| 2xWide-Resnet-50 | CIFAR | Wide_Resnet-50_2 | 68.9M | ImageNet | Yes | Yes | 0.0003 | 8.78 | 160.14 | -0.57 | 14.04 | 6 | 7.83 | 124.55 | 14.06 | 6.23 |
| Resnet-101 | CIFAR | Resnet-101 | 44.5M | ImageNet | Yes | Yes | 0.0003 | 10.91 | 166.44 | -0.62 | 14.97 | 5.03 | 8.95 | 117.22 | 14.81 | 5.86 |
| 2xWide-Resnet-101 | CIFAR | Wide_Resnet-101_2 | 126.9M | ImageNet | Yes | Yes | 0.0003 | 7.56 | 116.21 | -0.5 | 13.37 | 5.81 | 7.55 | 116.26 | 13.36 | 5.81 |
| **WIDTH_LINEAR** | | | | | | | | | | | | | | | | |
| Resnet-50 | CIFAR | Resnet-50 | 205K | ImageNet | No | Yes | 0.0003 | 13.5 | 56.54 | -0.22 | 28.63 | 6.66 | 23.05 | 112.08 | 28.65 | 5.6 |
| 2xWide-Resnet-50 | CIFAR | Wide_Resnet-50_2 | 205K | ImageNet | No | Yes | 0.0001 | 21.78 | 56.88 | -0.35 | 28.77 | 4.89 | 24.45 | 87.19 | 28.81 | 4.36 |
| Resnet-101 | CIFAR | Resnet-101 | 205K | ImageNet | No | Yes | 0.0003 | 11.17 | 51.7 | -0.21 | 25.86 | 6.17 | 20.98 | 99.32 | 25.95 | 4.97 |
| 2xWide-Resnet-101 | CIFAR | Wide_Resnet-101_2 | 205K | ImageNet | No | Yes | 0.0003 | 16.95 | 48.35 | -0.25 | 27.76 | 5.41 | 23.27 | 90.85 | 27.81 | 4.54 |
| **AUG_NO_PRETR_FT** | | | | | | | | | | | | | | | | |
| Resnet-18 w/ Data-Aug | Places | Resnet-18 | 11.7M | Random | Yes | Yes | 0.001 | 35.79 | 118.94 | -0.28 | 58.01 | 12.44 | 44.39 | 263.63 | 57.57 | 13.18 |
| Resnet-18 w/o Data-Aug | Places | Resnet-18 | 11.7M | Random | Yes | No | 0.003 | 36.62 | 114.2 | -0.24 | 63.73 | 13.01 | 48.84 | 288.46 | 63.26 | 14.42 |
| **AUG_PRETR_FT** | | | | | | | | | | | | | | | | |
| Resnet-18 w/ Data-Aug | Places | Resnet-18 | 11.7M | ImageNet | Yes | Yes | 0.0003 | 35.72 | 67.85 | -0.22 | 53.88 | 7.99 | 44.82 | 174.69 | 53.55 | 8.73 |
| Resnet-18 w/o Data-Aug | Places | Resnet-18 | 11.7M | ImageNet | Yes | No | 0.001 | 39.43 | 68.44 | -0.23 | 56.68 | 7.94 | 47.34 | 183.99 | 56.53 | 9.2 |
| **AUG_PRETR_LINEAR** | | | | | | | | | | | | | | | | |
| Resnet-18 w/ Data-Aug | Places | Resnet-18 | 187K | ImageNet | No | Yes | 0.0003 | 55.55 | 58.56 | -0.43 | 60.01 | 3.83 | 56.11 | 76.95 | 59.95 | 3.85 |
| Resnet-18 w/o Data-Aug | Places | Resnet-18 | 187K | ImageNet | No | No | 0.001 | 54.2 | 54.27 | -0.36 | 60.48 | 4.52 | 55.62 | 96.08 | 60.42 | 4.8 |
| **ARCHITECTURES_IN2CIFAR** | | | | | | | | | | | | | | | | |
| AlexNet | CIFAR | AlexNet | 61.1M | ImageNet | Yes | Yes | 0.001 | 7.52 | 131.77 | -0.32 | 26.89 | 12.4 | 15.97 | 219.36 | 26.94 | 10.97 |
| VGG-16(bn) | CIFAR | VGG-16BN | 138.4M | ImageNet | Yes | Yes | 0.0003 | 6.21 | 125.57 | -0.38 | 19.1 | 9.79 | 10.07 | 181.1 | 19.13 | 9.06 |
| ResNet-50 | CIFAR | Resnet-50 | 25.6M | ImageNet | Yes | Yes | 0.001 | 15.11 | 178.61 | -0.67 | 18.33 | 4.32 | 13.3 | 99.88 | 18.29 | 4.99 |
| ResNeXt-50(32x4d) | CIFAR | ResNeXt-50(32x4d) | 25.0M | ImageNet | Yes | Yes | 0.001 | 12.67 | 185.51 | -0.65 | 16.45 | 4.91 | 11.37 | 102.91 | 16.52 | 5.15 |
| ResNet-101 | CIFAR | Resnet-101 | 44.5M | ImageNet | Yes | Yes | 0.0003 | 10.91 | 166.44 | -0.62 | 14.97 | 5.03 | 8.95 | 117.22 | 14.81 | 5.86 |
| **ENSEMBLE** | | | | | | | | | | | | | | | | |
| 1xResnet-18 | CIFAR | Resnet-18 | 11.7M | ImageNet | Yes | Yes | 0.001 | 12.48 | 194.19 | -0.57 | 18.86 | 7.28 | 11.37 | 150.73 | 18.91 | 7.54 |
| 6xResnet-18 | CIFAR | Resnet-18 | 70.1M | ImageNet | Yes | Yes | 0.001 | 8.73 | 136.26 | -0.49 | 15.96 | 7.09 | 8.55 | 147.16 | 15.91 | 7.36 |
| 1xResnet-50 | CIFAR | Resnet-50 | 25.6M | ImageNet | Yes | Yes | 0.001 | 15.11 | 178.61 | -0.67 | 18.33 | 4.32 | 13.3 | 99.88 | 18.29 | 4.99 |
| 3xResnet-50 | CIFAR | Resnet-50 | 76.7M | ImageNet | Yes | Yes | 0.001 | 12.72 | 150.13 | -0.64 | 15.96 | 4.15 | 11.43 | 90.59 | 15.96 | 4.53 |

# D    DRAFT ADDITIONAL CHANGES TO INCLUDE IN FINAL VERSION

This section contains preliminary results and text requested by reviewers that will be carefully integrated into the main document in final revision.

## D.1    OPTIMIZATION EXPERIMENTS

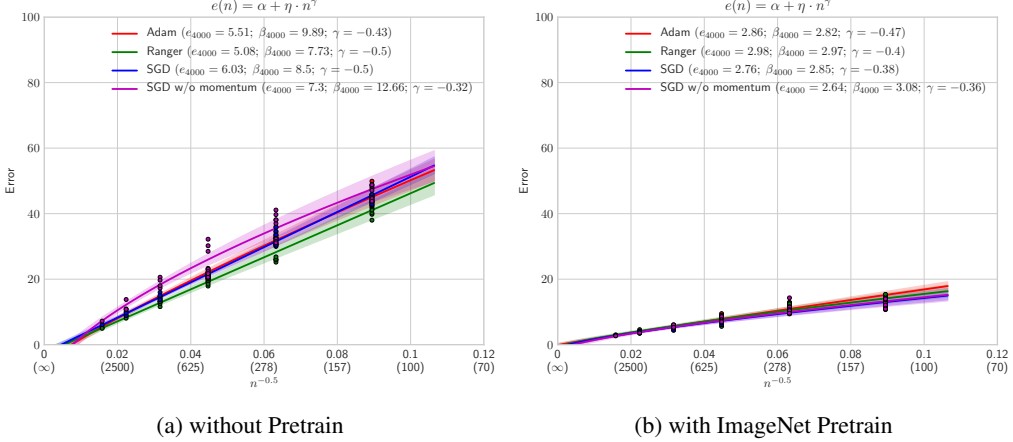

(a) without Pretrain                                      (b) with ImageNet Pretrain

Figure 9: **Optimization** on Cifar10 with ResNet-18.

In Fig. 9, we show results on Cifar10 when training ResNet-18 using four different optimization methods: Ranger (Wright, 2019), Adam (Kingma & Ba, 2015), stochastic gradient descent (SGD) w/ momentum, and SGD w/o momentum. Similarly to our experiments with Cifar100, we use 80% of the standard training set for training and validation (4000 examples per class) and the remaining 20% for testing. With pretraining, all methods perform similarly, but when training from scratch (no pretraining), Ranger outperforms with lower $e_{4000}$ and $\beta_{4000}$. SGD without momentum performs the worst and is least consistent across folds.

## D.2    TEXT FOR SECTION 2.2

However, if the classifier parameters are functions of $n$, then $\gamma$ may deviate from $-0.5$. For example, Tsybakov (2008) shows that a kernel density estimator (KDE) with fixed bandwidth $h$ has MSE bounded by $O(\frac{1}{nh})$, but when the bandwidth is set as a function of $n$ to minimize MSE, the bound becomes $O(n^{-\frac{2\beta}{2\beta+1}})$ where $\beta$ is the kernel order. In our experiments, all aspects of our model are fixed across training size when estimating one learning curve, except learning schedule, but it should be noted that error bounds and likely the learning curve parameters depend on both the classifier form and which parameters vary with $n$.

## D.3    OTHER PLANNED IMPROVEMENTS

- Experiments to include WRN-28-10 (or similarly effective Wide ResNet model) on Cifar-100 to show that learning curve methodology applies and experimental findings hold for high-performing models

- Discussion to clarify that experiments serve to exemplify use of learning curves and make interesting observations, but more extensive study of each design parameter is warranted. Also discuss any other concerns/limitations raised by reviewers.

