# OpenReview forum: "Learning Curves for Analysis of Deep Networks"
_ICLR.cc/2021/Conference — Reject_

### Official Review · AnonReviewer4 · 2020-10-25

**Rating:** 6
**Confidence:** 4

**Review:**

This paper advocates for studying the effect of design choices in deep learning via their effect on entire *learning curves* (test error vs num samples N), as opposed to their effect only for a fixed N. This is a valid and important message, and it is indeed an aspect that is often overlooked in certain domains. However, although this paper addresses an important issue, there are methodological concerns (described below) which prevent me from recommending acceptance. In summary, the paper oversimplifies certain important aspects in both the setup and the experiments.

Concerns:
1. My main concern is that the discussion of learning curves ignores the effect of model size. Prior work (including Kaplan 2020 and Rosenfeld 2020) has shown that learning curves exhibit quantitatively different behavior when models are overparameterized vs. underparameterized. In particular, learning curves are only known to exhibit clean power-law behavior when model-size is not the bottleneck (e.g. if model-size is scaled up correspondingly to data size). There is no discussion of the model-size issue in the present work. This may be problematic, since data from small N are used to extrapolate to large N, but the model size is held fixed.
Concretely: a full discussion of how to evaluate and interpret learning curves should account for the effect of model-size.

2. The curve-fitting procedure is non-standard, and produces some questionable extrapolations.
This is concerning because one of the stated contributions of this paper is to propose an experimental methodology. Specifically:

A. If the true parametric form is a power-law with some \gamma != 0.5, why are the learning curves plotted assuming \gamma=0.5 (Figure 1)? In the regression estimate (Equation 7), why is the exponent \gamma encouraged to be close to 0.5?
Note that the theoretical justification for \gamma=0.5 (Table 2) is weak -- it only includes parametric bounds. Non-parametric rates are in general different from \gamma=0.5.


B. Several of the curves in Figure 1 predict cross-overs which we do not expect to occur at N=infty. For example, Figure 1g predicts that an ensemble of 6 ResNet18s will be better than 1 ResNet50 at N=\infty, which we do not expect.

C. In general, the curves are extrapolated from only 5 points -- it would be more convincing to see more data-sizes tested.

3. Regarding experimental setup and conclusions:

A. Why are there experiments for CIFAR-100 but not CIFAR-10? Most of the current experiments have high error rates (~20%), so it would have been nice to see how the curve-fits perform down to low error-rates (< 5%) as we would see on CIFAR-10.

B. The claim that "pretraining does not bias the classifier" is too strong to be supported by the experiments. Certainly this does not hold for any arbitrary pre-training dataset, but perhaps it holds for "natural" pre-training datasets close to the ones tested here. In general, several of the experimental claims are too strong in this way -- they make universal statements, but are only tested in a few limited respects. Further experiments would give more evidence to these claims. For example, it is speculated on pg 11 that \gamma does not depend much on the model architecture. Does this continue to hold for MLPs? (Only convnets are tested in this paper).


Summary: The motivation of this paper is very good, but the proposed experimental methodology is somewhat lacking. This paper would be much improved by more thorough experiments and analysis, and more nuanced discussion of the experimental conclusion.


Comments/clarifications which do not affect the score:
Why are the experiments done using the Ranger optimizer? Would any conclusions differ if we use standard optimizers (SGD/Adam)?
I would suggest moving Section 3.2 to the appendix, since the mechanics of least squares is likely familiar to readers. This would open more space for further discussion of Figure 1 experiments.

---
Edit after rebuttal: Changed score from 5 to 6 (see below)

---

> ### Author Response · Authors · 2020-11-16
> **Re to AnonReviewer4 (model size interactions, clarification about gamma fitting, gamma prior, extrapolations, others)**
>
> [main concern is that discussion ignores the effect of model size]
> We do discuss Kaplan et al. 2020 and Rosenfeld et al. 2020 in Sec. 2.2 and bring them up three more times in the extended discussion in Appendix A. So we do not think it is accurate to say “there is no discussion of the model-size issue”, but we take your point that it is an important consideration and have added another mention in the experiments section as well.
>
> Rosenfeld et al. shows that error can be modeled as a function of model size or data size or both jointly. Joint accounting is helpful for some purposes (e.g. selecting model size from data size or vice versa), but this requires a large number of experiments.  We show how to compute learning curves and characterize them in a way that is stable given errors from only a few data sizes. Appendix C shows that e_N and beta_N are stable even when estimating from as few as three data sizes, which makes our learning curve method usable by researchers with modest computational budgets.
>
> [why are learning curves plotted assuming gamma=-0.5]
> We estimate gamma for each curve, as described in comments to Anon3, and have revised the text to make this more clear. The gamma values are shown in the legends.
>
> The main reason that we use a prior for gamma=-0.5 is to reduce variance on the individual parameters without substantially changing the path of the curve in the observed range. Eta and gamma are highly covariant (e.g. the derivative of error with n includes a product of gamma and eta). Our characterization in terms of data-reliance beta reduces the importance of this covariance, but the prior still makes sense because we observe that when the curve fits best and variance per data size is small, gamma tends to be estimated close to -0.5 without using a prior.  Likewise, Rosenfeld et al. 2020 independently find values of gamma between -0.5 to -0.6 for multiple image classification tasks in a large set of experiments. In most cases, the prior does not have much effect, except when there are few error measurements.
>
> [large extrapolations may not be accurate]
> Our experiments show that extrapolations to 4x the data size are accurate, but we caution against larger extrapolations, especially to n=infinity.  This is stated under the limitations and indicated by the bar at n=1600 in the figures, as discussed in the beginning of Sec. 4.2. This means that the learning curves cannot tell us what would happen with infinite data (which would be hard to validate), but they can still be useful for characterizing performance and estimating impact of doubling or quadrupling training size.
>
> [use more data sizes]
> We agree that more analysis on large datasets would be interesting. Our range of data sizes is limited by the size of the dataset for CIFAR-100. We show that the curves achieve good leave-one-out error prediction with even a few data sizes.  This was by design, as we want researchers to be able to compute learning curves on a modest computational budget, so that they can be used widely. We compute errors for multiple data splits at each training size to enable more accurate fits and to measure variance.
>
> [why not use CIFAR-10]
> We had previously not prioritized Cifar-10 because of its similarity to Cifar-100, but we are currently running some experiments because you are right that Cifar-10 affords a bigger range of data sizes for analysis. We do see much lower error rates for Caltech-101 and Pets, in the range of 5-10% (now “Fig 6: additional datasets” in revised version).  In our experiments so far, the ResNet-18 with Ranger achieves 5.6% error with 4000 training samples per class on Cifar-10 without pretraining, and 3.4% error with pre-training.
>
> [“pretraining does not bias classifier” claim is too strong]
> Thank you for pointing this out. We should not have made a claim that required knowledge of asymptotic performance, and we have revised the text to be more accurate in this regard. We agree that more experimentation is merited to explore some of these problems.  As we responded to AnonReviewer2, we felt that a broad set of experiments would best highlight the value and broad applicability of using learning curves for analysis, but much more work is needed to fully explore some of these design decisions.
>
> [why Ranger? Dependence on optimization?]
> In preliminary experiments, we found Ranger to provide lower error rates and less sensitivity to hyperparameters than Adam or vanilla SGD. The hyperparameter sensitivity issue was particularly important to us, given our broad range of experiments. We’ve clarified this in the text. We are running experiments to compare to SGD/Adam on CIFAR-10 to address two of your comments at once, but are not sure if they will be done in time for discussion phase.
>
>  [move 3.2 to appendix]
> This is a good idea, but we are able to expand discussion of experiments in revision without moving it, because ICLR allows an extra page for discussion phase and camera ready.

---

> > ### Comment · AnonReviewer4 · 2020-11-16
> > **response**
> >
> > Thank you for the discussion so far. Are the authors planning on including a discussion on non-parametric rates in Section 2.1, as mentioned in my review? It should be noted that there are generalization bounds with exponents different from gamma=0.5 in the literature, for non-parametric estimators.
> >
> > See for example Theorem 1.1 in "Introduction to Nonparametric Estimation" by Tsybakov. (And also similar theorems in later chapters of the same text).

---

> > > ### Author Response · Authors · 2020-11-17
> > > **gamma when classifier (hyper)parameters depend on n**
> > >
> > > Thank you for this quick follow-up.  We did not fully get your meaning before, but this is an important point.  If model parameters, such as depth and width, and maybe even learning rate and regularization parameters, are allowed to vary as a function of $n$, then theoretically $\gamma$ may not be -0.5.  We propose adding to the end of Sec 2.1:
> > >
> > > However, if the classifier parameters are functions of $n$, then $\gamma$ may deviate from $-0.5$. For example, Tsybakov (2008) shows that a kernel density estimator (KDE) with fixed bandwidth $h$ has MSE bounded by $O(\frac{1}{nh})$, but when the bandwidth is set as a function of $n$ to minimize MSE, the bound becomes $O(n^{-\frac{2\beta}{2\beta+1}})$ where $\beta$ is the kernel order.  In our experiments, all aspects of our model are fixed across training size when estimating one learning curve, except learning schedule, but it should be noted that error bounds and likely the learning curve parameters depend on both the classifier form and which parameters vary with $n$.
> > >
> > > If you don't mind, we will wait to upload a revised version with this change because small changes mess up the formatting with wrapfigure, which requires a lot of tweaking to resolve, but we have checked that the text with this included easily fits within the page limit.
> > >
> > > Does this address your comment?

---

> > > > ### Comment · AnonReviewer4 · 2020-11-21
> > > > **response**
> > > >
> > > > Yes, adding that paragraph would address my concerns about non-parametric rates.
> > > >
> > > > I have increased my score to 'weak accept', acting on the assumption that the additional experiments you mention will be included in the final version.
> > > > I believe 'weak accept' is now justified, since in the revision, Section 4.2 is now just presented as examples of how to use the methodology (and not absolute conclusions, which would require more justification).
> > > >
> > > > The main point of this paper is to argue that practitioners should be using entire learning-curves to evaluate design choices (instead of just reporting accuracies for the full dataset). This is a good point, and the paper takes steps to establish such a methodology. The paper includes several examples of using this methodology, though it should be seen as just that -- **examples**, which should encourage practitioners to conduct similar studies (but more thorough) when publishing future papers. In light of this, I believe the paper should be accepted, because it takes another step in encouraging more careful evaluation of learning procedures -- and the revised experimental section addressed most of my concerns.
> > > >
> > > > Comments which do not affect the score:
> > > > I encourage the authors to include all of the concerns of the reviewers in the "limitations" section of the final version, and explain them clearly. As well as a more thorough discussion of the objective of this work, in comparison to other works on scaling laws. It will only make the paper stronger and better-received.

---

> > > > > ### Author Response · Authors · 2020-11-21
> > > > > **re: response**
> > > > >
> > > > > Thank you once again for all of your comments, which have been very helpful in improving the paper.  Also, thank you for adjusting your rating.  We agree with your summary of contributions and are glad that you find them worth acceptance.
> > > > >
> > > > > The additional experiments on optimization will be included, and we will also further reflect review discussions in the final version.

---

### Official Review · AnonReviewer1 · 2020-10-29

**Rating:** 7
**Confidence:** 3

**Review:**

### Summary
The authors conduct an investigation of learning curves on image classification models to understand tradeoffs between error and dataset size across different design choices. These learning curves help clarify the relationship between error and data reliance as a function of choices such as depth, width, pre-training and fine-tuning.

### Comments
* The introduction and motivation for the work is clear and well written.
* I found the work well-contextualized within existing work.
* The learning curve is a nice way to understand tradeoffs in design choices for a given model.
* A single choice of optimizer is used, and learning curves could conceivably vary between different optimizers. It would be good to explore this further.
* The experiments are thorough and have interesting conclusions that are applicable to researchers / practitioners e.g. deeper networks are suitable for smaller datasets.

### Recommendation / Justification
I vote to accept the paper. The authors do an excellent job of motivating the importance of learning curves and are systematic about their experimentation and analysis.

### Minor feedback
* Missing header and page number on first page
* I like the T/F quiz. I think it'd help to put more detailed descriptions of some of the procedures e.g. "Fine-tuning the entire network is only helpful if the training set is large" is vague if you do not quantify helpful nor define an alternative (such as fine-tuning just the final layer).
* typo:  "pseudeo-inverse"
* appendix discussion on "impact of \gamma" $\gamma = 0 - 0.5$ would probably be clearer as $\gamma \in (0, 0.5)$

---

> ### Author Response · Authors · 2020-11-16
> **detailed response to AnonReviewer 1 (choice of optimizer experiments)**
>
> [would be good to explore choice of optimizer further]
> We agree. We are running experiments comparing SGD (with and without momentum), Adam, and Ranger now on CIFAR10, but we are not sure whether they will be ready within the rebuttal period. In general, the effect of hyperparameters and optimizers on learning curves is an important area to explore, both to aid understanding and establish a best practice for training.
>
> [minor feedback]
> Thanks for these suggestions and corrections; we’ve done our best to address in revision.

---

### Official Review · AnonReviewer3 · 2020-10-29
**A simple, easily usable method to predict learning curves and data reliance, with interesting insights on transfer and data augmentation**

**Rating:** 7
**Confidence:** 4

**Review:**

In this paper, the authors first propose a simple weighted least squares method to compute the "learning curve" (error plotted against dataset size) , where error is modelled with the form error = alpha - eta*n^gamma, for parameters alpha, eta, gamma. Gamma is taken to be - 0.5, while alpha, eta are estimated from the data. This also allows an estimate of "data reliance", in essence the slope of error wrt dataset size, computing how much error decrease is dependent on dataset size.

The authors then perform an extensive experimental evaluation on varying sized subsets of CIFAR-100 and Places365, across multiple different neural architectures and varying choices of finetuning, pretraining, linear classifier (frozen feature training) varying architecture size and data augmentation. They fit learning curves on these different empirical configurations, estimating data-reliance (along with extrapolating error), finding interesting conclusions such as finetuning outperforming linear classifiers even on small datasets, and larger architectures actually improving data reliance even in small data settings.

Both the learning curve computations and empirical evaluations are interesting and I recommend accepting this paper.

However, I would strongly suggest changing the layout of the paper, in particular, splitting up Figure 1 into several figures to better emphasize some of the different takeaways. (E.g. perhaps section 3.2, which consists of recapping weighted least squares could be moved to the supplementary.) As it is, it is very difficult to follow the main takeaways from the different experiments, and even the insights given by fitting the learning curve (and computing data-reliance.)

If the authors can provide such a revised version, I would definitely consider further increasing my score.

Minor comments:

-- I appreciate the open acknowledgement of some of the limitations of the method

-- I also liked the summary table (deep learning quiz) summarizing some of the conclusions of Figure 1 (which would have been hard to absorb otherwise)

-- Are there assumptions about the dataset/task that must be made for fitting learning curves (and predicting data reliance) to work well? For example, what if the model is trained to memorize random labels?

---

> ### Author Response · Authors · 2020-11-16
> **detailed comments to AnonReviewer3 (clarification about fitting gamma, split fig 1, assumptions)**
>
> [Gamma is taken to be -0.5]
> Note that gamma is also estimated from the data, and the estimated value is shown for each curve in the legend.  The x-axis scale is in n^(-0.5), and beta_400 is based on the derivative with respect to n^(-0.5), the learning curve and beta are still based on the estimated gamma.  We revised the text to make this more clear.
>
> [Split Figure 1 into several figures]
> Great idea! We previously condensed the figures to save space (maybe would have been better to move 3.2 as you suggest), but as we are now allowed an extra page, the revision better distributes the plots in the text, and this does make the experiments section much easier to read.  We also expand the discussion of results a little.
>
> [assumptions about dataset/task?]
> We haven’t experimented with other classifiers, but we expect learning curves would apply widely, as the parameterization is in part based on general principles such as bias-variance decomposition of error.  A classifier trained on random labels would simply have asymptotic error of chance and an eta coefficient of 0. The main limitation is likely the evaluation metric. While learning curves apply to logistic loss and classification error, the parametric model might not apply to AP, ROC area, segmentation IoU, Bleu score, or other task-specific metrics.

---

> > ### Comment · AnonReviewer3 · 2020-11-24
> > **Response to Authors**
> >
> > Thank you to the authors for their detailed responses and the revised version of the paper, which, with the split figure, makes it much easier to appreciate the conclusions.
> >
> > I recommend accepting the paper, but am keeping my score as is, in light of the comments made by reviewer2, on trying out additional models, e.g. WideResNet-28-10 that give better task performance.
> >
> > I think the insights from this paper will be of broad interest to the community, but particularly when studying properties of transfer/data augmentation, it's always helpful to see the results on a broad range of models (even if there is some variation in the results.)

---

### Official Review · AnonReviewer2 · 2020-10-29
**The conclusions reached by the authors are not justified by their experiments.**

**Rating:** 4
**Confidence:** 3

**Review:**

This paper uses experimental measurements and empirical curve-fits of learning curves to study their interaction with training protocols such as transfer learning and data augmentation. They define the learning curve to be the test error as a function of training set size.

I appreciate the authors' goal of relating the scaling behavior of learning curves with common choices in NN training. Deep learning models are often complex and hard to model from first principles. Trying to understand and design deep learning models using scaling laws, empirical measurements, and power-law fits seems like a great idea. However, I am not entirely sure if the conclusions of this paper are convincing enough to be sufficiently useful to the ICLR readership. For example:

a) The authors conclude that "Pre-training on similar domains nearly always helps compared to training from scratch.", in agreement with their initial guess before they ran their experiments.  As far as I can tell, they reached this conclusion because one model they plot in Figure 1d, which was trained from random initializations, does worse on average than the 3 models they finetuned, all of which were pre-trained on larger datasets. It has already been reported in literature that pre-training on a similar dataset does not always outperform training from scratch [1 (already cited), 2, 3]. In fact, Ref. [3] has shown that depending on the data-augmentation settings, pre-training can significantly hurt final performance compared to training from scratch. Furthermore, Ref. 2 found that pre-training will not be helpful for certain dataset pairs, even if the target dataset has "similar" classes as the source dataset (e.g. cars in FGVC cars and ImageNet). I find the conclusion from Ref. 2 more convincing and precise, since they have evaluated transferring to 12 different datasets, using 16 different architectures, compared to the 2 different target datasets in this paper, and only 1 architecture (authors mention that they trained 8 different architectures, but am I understanding correctly that Figure 1d only includes a single architecture that is not pre-trained?)

b) Building on item (a), I am not sure if the model trained without pre-training is representative of what researchers would use. Accuracies are not reported in the text, but it reads as though the "No Pretr" model achieves around 72% accuracy on CIFAR-100 (Fig 1d). It is relatively easy to achieve above 80% with a standard architecture such as WideResNet-28-10. Other work cited above have already seen that pre-training is more likely to do better than models trained from scratch if the models trained from scratch are not trained as optimally as they could be (common culprits are not training for long enough, not using sufficient regularization etc.) For this reason, in order to claim something as strong as  Popular Beliefs #1, it is important to have a good baseline for the model trained from scratch (note that I am not asking for a state-of-the-art results here, but a result that is comparable to models trained in literature). (One reason could potentially be that the authors trained on 40k samples of CIFAR-100 instead of the commonly used 45k, but it is still easy to get 80%+ with a standard training protocol initialized to random weights on 40k samples of CIFAR-100).

c)  I worry that similar issues exist for other conclusions of the paper. For example, the conclusions about "Increasing the network depth" are very interesting, but I am not convinced that the paper has the experiments to justify them. It is crucial to optimize the regularization techniques for each width and depth separately, before one can make a statement about whether increasing the depth or width is generally helpful or not.

I hope that the authors continue this work, expand and improve their experimental setup. Relating learning-curves to models that are of interest to the community, with performance that match what is reported in literature, would be a great first step. This way, the readers can easily judge if the connections observed in the paper would be applicable to the standard training protocols. A next step could be to show that using the insights that are developed in this paper, the authors can achieve a result that was not possible without their insights.

[1] He, Kaiming, Ross Girshick, and Piotr Dollár. "Rethinking imagenet pre-training." Proceedings of the IEEE international conference on computer vision. 2019.
[2] Kornblith, Simon, Jonathon Shlens, and Quoc V. Le. "Do better imagenet models transfer better?." Proceedings of the IEEE conference on computer vision and pattern recognition. 2019.
[3] Zoph, Barret, et al. "Rethinking pre-training and self-training." arXiv preprint arXiv:2006.06882 (2020).

---

> ### Author Response · Authors · 2020-11-16
> **detailed response to AnonReviewer2**
>
> [“pretraining on similar domains nearly always helps” conclusion is too strong or not given enough context]
> Thank you for these comments and additional references. We have updated the pretraining paragraph in experiments and clarified that T/F in the “quiz” is meant to indicate whether our experimental results are consistent with each belief.  We also discuss the references you cited.  Note that [2] (your second reference) also finds that pre-training outperforms random initialization in nearly all cases in their scatter plot, but the gap is small and shrinking with increased data size.
>
> Our experiments are intended to provide some interesting observations, demonstrate that learning curves are useful for studying a wide variety of design decisions, and indicate areas for further exploration. As such, we explore several design choices lightly, but do not intend to claim that our experiments are comprehensive or provide the final word in any single area, as that would require several papers worth of exploration.  We try to make this clear in our discussion of limitations in the main paper and appendix.
>
> [concern about accuracy and architecture]
> Our pretrained ResNet-101 model achieves about 85% accuracy (e_400=14.97) on CIFAR100, and we can use eq 1 and our learning curve parameters to estimate that the accuracy with 500 samples per class (standard train set) would be 85.6%.  The same analysis on ResNet-18 without pretraining (1a) estimates accuracy of 72.1% with 400 samples per class or 73.0% with the full set.
>
> The best reported accuracy on CIFAR100 with a Resnet-18 is 78.3% with Lookahead according to paperswithcode.com/sota/stochastic-optimization-on-cifar-100-resnet (Zhang et al. NeurIPS 2019).  But Zhang et al. note (Appendix C) that their network is wider and uses more parameters than standard ResNet-18, and they are able to tune hyperparameters and learning schedules to maximize test accuracy, while we optimize only learning rate and number of epochs using a subset of training as validation (mini-val).  Given these considerations, our performance seems in the same ballpark as state-of-the-art.
>
> [could parameters have been tuned better]
> As we note in limitations, the learning curves are likely dependent on optimization and hyperparameters, but as noted above, it seems that our models were reasonably well trained. We do search for learning rate for each model and learning schedule for each data size using our mini-val set, which we find important for achieving well-fitting curves and minimizing variance for different data splits for fixed model and training size.
>
> [relate learning curves to models of interest to community with performance that matches literature]
> As above, we believe our training practice is comparable to the literature best practice, and our experiments feature commonly used models and the latest optimizer Ranger that reduces sensitivity to hyperparameters.  Our aim is not to achieve the best result, but to show that the learning curves can provide a useful analysis. We believe a methodology like this is needed because many papers either neglect to report effect of training size or do so in an ad hoc way, e.g. by also reporting performance on some percent of training size.  It is difficult to propose a new evaluation methodology at the same time as performing a survey or making an algorithmic contribution.  By thinking through the decisions of how to compute, characterize, and display learning curves, we make it easy for others to use learning curves to better understand their own contributions or explore particular design decisions in detail.

---

> > ### Comment · AnonReviewer2 · 2020-11-21
> > **response**
> >
> > Thank you for the response and the clarifications. I unfortunately do not agree with your statement that "our training practice is comparable to the literature best practice". I think that further experimental diversity and rigor is needed for an empirical study like yours. Below I respond in detail:
> >
> > > Note that [2] (your second reference) also finds that pre-training outperforms random initialization in nearly all cases
> >
> > The results in [2] show that pretraining is nuanced, and it is difficult to support general statements such as "Pre-training on similar domains [...] nearly always helps" with experiments done only on a single model and 2 datasets. The authors in [2] found that for 3 models finetuned on FGVC aircraft, the finetuning accuracy was lower than a random initialization. The authors stated that "on Stanford Cars and FGVC Aircraft, the improvement was unexpectedly small", even though Stanford Cars is a "similar domain" to ImageNet.
> >
> > > The best reported accuracy on CIFAR100 with a Resnet-18 is 78.3% with Lookahead according to paperswithcode.com/sota/stochastic-optimization-on-cifar-100-resnet (Zhang et al. NeurIPS 2019).
> >
> > 5.3% difference with what's reported in literature is not insignificant. Furthermore, as I mentioned in my original review, one can train a wide-resnet-28-10 model to 82% accuracy in a few hours on a single GPU, which is a commonly used architecture for this dataset.
> >
> > > Our aim is not to achieve the best result
> >
> > I wholeheartedly agree that you do not have to aim to achieve the best result. Best result on this dataset is 90%+. I was certainly not suggesting you should train such a model. However, there is a large gap between 90% and 72%. Given the general availability and low computational cost of models that achieve 82%+ with a random initialization, I am not sure one can justify using only ResNet-18 to make a such general statements about the general efficacy of finetuning on Cifar-100. By reporting better models that are more commonly studied in literature, it will be possible for the readers to directly relate their experiments to your results.
> >
> > Another comment is that I like the new layout of the figures, but I'm curious if it should be clear to the reader which dataset the models on Figure 3 were finetuned on? I see that Figure 2a is for Cifar100, 2b is for Places365, which one is Figure 3 for?

---

> > > ### Author Response · Authors · 2020-11-21
> > > **re: Response**
> > >
> > > Thank you very much for your additional response.  Your comments have already led to improvements in the paper, and we really appreciate that you've taken this time and thank you for the opportunity to further respond.
> > >
> > > ### Claims regarding effectiveness of pre-training
> > >
> > > After your first response, we revised the experimental discussion to include more of the literature, including the references that you mentioned. We also clarified that the "T" in the results column of the "deep learning quiz" indicates whether our experimental results are consistent with a belief.  In the case of pre-training, we agree that other papers have a more complete exploration of the topic. We find the comparison of pre-training sources to be interesting, but our main objective is to give examples of how learning curves can be useful, and this is just one of several design parameters that we investigate.
> > >
> > > When we said that [2] supports that pre-training nearly always helps, we are referring to Fig E.1 in that paper, where they compare random initialization to pretraining on 12 datasets and 16 architectures.  Out of the 192 cases, only three result in slightly better accuracy for random initialization, so in 98.5% of their experiments, pretraining helped, including 13 out of 16 of the FGVC cases.
> > >
> > > ### Fig 3 -- which dataset is used
> > >
> > > We are glad the new layout is helpful.  Fig 3 and 4 test on Cifar-100.  We will clarify in revision. Thank you for pointing this out, and we'll make sure it's very clear in the revised version.
> > >
> > > ### Concerning Cifar-100 accuracy and whether there is a flaw in our training
> > >
> > > We agree with you that 5.3% difference in error is substantial, but it's an "apples and oranges" comparison.
> > >
> > > * Method 1 (ours): Standard ResNet-18. Perform training and hyperparameter search, including stopping iteration, using the 80% of the training set. Test on fully held-out data.  73% accuracy.
> > > * Method 2 (theirs): Modified ResNet-18 with more parameters. Use 100% of training data for learning weights and use test error to choose hyperparameters and when to stop based on test error.  Report the best of three runs.  78% accuracy.
> > >
> > > It does not seem unreasonable to say that these differences could be explained by that they tune meta-parameters on the test error, and we do not.  Unfortunately, it is common practice in the literature to tune meta-parameters based on the test error for Cifar-100, which makes it difficult to use those experiments as a basis for determining whether we have a problem with our training regime.
> > >
> > > Looking at Places365, the dataset creators report 45% error using ResNet-152 trained on 1.8 million images, while we report roughly 50% error using ResNet-18 trained on 584,000 examples.  Again, it's not an exact comparison, but it does not seem to indicate a problem with our experiments.
> > >
> > > **In summary**, we have made a reasonable effort to train well using standard tools and believe that differences in architectures, data augmentation, and especially the prevalent practice of using test data to tune meta-parameters could account for differences in reported numbers. We do not think these differences indicate a flaw in our training method.
> > >
> > > We do agree that a more in-depth exploration of any particular topic such as data augmentation, effects of model size, ensembles, and pre-training should include experiments on several architectures.  It's hard to do this for all of them at once though, and we went broad rather than deep to show the broad applicability for learning curves and to discover areas in need of deeper exploration.

---

> > > > ### Comment · AnonReviewer1 · 2020-11-23
> > > > **Comment on ResNet-18 Differences**
> > > >
> > > > I agree with the authors here that the differences in ResNet-18 performance should not be held against them. I believe the source of the discrepancies may be due to this repository: https://github.com/uoguelph-mlrg/Cutout
> > > > which the Lookahead paper references and achieves comparable accuracies to.  Unfortunately, it looks like this architecture deviates from that of the original ResNet-18 paper.
> > > >
> > > > >Given the general availability and low computational cost of models that achieve 82%+ with a random initialization, I am not sure one can justify using only ResNet-18 to make a such general statements about the general efficacy of finetuning on Cifar-100. By reporting better models that are more commonly studied in literature, it will be possible for the readers to directly relate their experiments to your results.
> > > >
> > > > However, I do agree with the reviewer2's point, which I missed in my original review, that running experiments with WRN-28-10 would strengthen the paper, as it is an established baseline which achieves stronger performance and lead to more confidence in the conclusions. I hope that such experiments would make the final revision of this paper.

---

> > > > > ### Author Response · Authors · 2020-11-23
> > > > > **add WRN-28-10 experiments**
> > > > >
> > > > > Yes, AnonReviewer2 and AnonReviewer1 make a good point that the paper will be improved by including experiments on WRN-28-10, since it serves as a stronger baseline architecture, and we will include experiments with WRN-28-10 in final revisions.

---

### Author Response · Authors · 2020-11-16
**Thank you to reviewers + summary of revisions**

Thank you to the reviewers for your thoughtful, informative, and constructive comments!  Each of your remarks has been helpful to improve the paper.

Our main changes are:
1. Corrected/clarified experimental discussion to incorporate reviewer comments
2. Clarify that gamma is not set to -0.5 but estimated for each curve
3. Break out the plots into multiple figures aligned with the corresponding text to make the experiments section much easier to read.
4. Expand text in some areas to discuss related works in the experiments, e.g. concerning model size and pre-training

We also respond individually with more detailed comments. We hope these revisions and comments will address most of the raised concerns. We would appreciate any further suggestions on how to improve the paper or increase likelihood of adoption.

---

### Author Response · Authors · 2020-11-24
**Updated revision, final thanks**

As the discussion period is now at its end, we again send our sincere thanks to the reviewers for their thoughtful comments and critiques, which  have led to substantial improvements in the paper.  Your informative reviews are a credit to ICLR.

We uploaded one more revision (see Appendix D on last page), which includes:
* experiments on optimization (Ranger outperforms when training from scratch, especially for smaller data sizes; all methods similar with pretraining)
* the paragraph to be added to Sec 2.2
* description of other planned changes (WRN-28-10 experiments, discussion)

These will be carefully integrated into the paper in final revision, but we put them in Appendix D for now to make them easier to find and also because we need more time to do a full proper revision.

We welcome any further suggestions.

---

### Decision · Program_Chairs · 2021-01-07
**Final Decision**

**Decision:**

Reject

**Comment:**

This paper studies the relationship between test error as a function of training set size and various design choices of neural network training. Overall all of the reviewers are excited about the prospect of relating error curves to neural network design choices, but different reviewers complain about the rigor of empirical evaluation and the accuracy of conclusions given limited data points. I agree with reviewers on both points, i.e., the paper studies different design choices, but does not do a thorough job studying those design choices. Moreover, it is not clear what aspects of the study are directly related to error curves vs. a standard correlation study done in prior work, e.g. in "Do better ImageNet models transfer better?" for usefulness of ImageNet pre-training. So, overall, I believe not only the empirical evaluation needs improvement, but also the story needs refinement. I am looking forward to seeing this paper published in other ML venues.